# One-shot Learning for Temporal Knowledge Graphs

**Mehrnoosh Mirtaheri**[1,2]                    MEHRNOOM@USC.EDU
**Mohammad Rostami**[1,2]                    MROSTAMI@ISI.EDU
**Xiang Ren**[1,2]                    XIANGREN@ISI.EDU
**Fred Morstatter**[1,2]                    FREDMORS@ISI.EDU
**Aram Galstyan**[1,2]                    GALSTYAN@ISI.EDU
[1]*University of Southern California,*
*Los Angeles, CA 90007 USA*
[2] *Information Sciences Institute,*
*Marina Del Rey, CA 90292 USA*

## Abstract

Most real-world knowledge graphs are characterized by a frequency distribution with a long-tail where a significant fraction of relations occurs only a handful of times. This observation has given rise to recent interest in low-shot learning methods that are able to generalize from only a few examples per relation. The existing approaches, however, are tailored to static knowledge graphs and do not easily generalize to temporal settings, where data scarcity poses even bigger problems, e.g., due to occurrence of new, previously unseen relations. We address this shortcoming by proposing a one-shot learning framework for link prediction in temporal knowledge graphs. Our proposed method employs a self-attention mechanism to effectively encode temporal interactions between entities, and a network to compute a similarity score between a given query and a (one-shot) example. Our experiments show that the proposed algorithm outperforms the state of the art baselines for two well-studied benchmarks while achieving significantly better performance for sparse relations.

## 1. Introduction

Knowledge graphs (KGs) provide a principled way of representing factual information and have become a crucial component for performing various Natural Language Processing (NLP) tasks, including cross-lingual translation [Wang et al., 2018], Q&A [Yao and Van Durme, 2014] and information retrieval [Dietz et al., 2018]. Semantic KGs such as YAGO [Kasneci et al., 2009] and WikiData [Vrandečić and Krötzsch, 2014] store *facts* as collections of triples in the form of (subject entity, relation, object entity), while Temporal Knowledge Graphs (TKGs) contain *events* presented as quadruples (subject entity, relation, object entity, t), where $t$ is a time-labeling of the edges, that associates punctual dates to the occurrence of the interactions between the entities. Despite development of advanced extraction techniques, KGs are typically incomplete, which limits the performance and range of KG-based applications. Recent research has focused on developing methods for predicting missing facts/events for static KGs [Trouillon et al., 2016, Xiong et al., 2017, Chen et al., 2018] and TKGs [Garcia-Duran et al., 2018, Xu et al., 2019, Wu et al., 2020].

Most existing KG completion methods rely on a sufficiently large number of training examples per relation. However, most real-world KGs have a long-tail structure, i.e, many relations occur only a handful of times. The data scarcity issue is exacerbated for temporal graphs, since the dynamics governing the evolution of those graphs might be highly non-stationary. First, new types

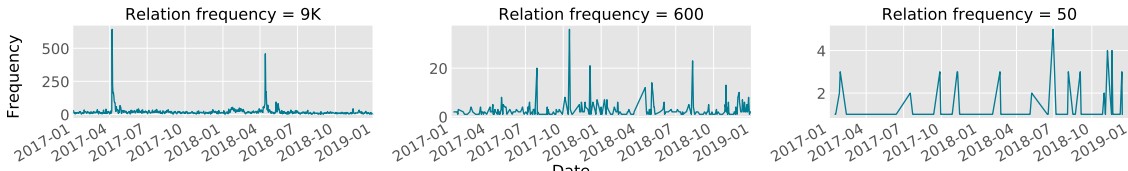

Figure 1: The highly heterogeneous distribution of occurrences for three relations with different frequency over ICEWS Jan 2017-Jan 2019.

of relations/events might emerge that have not been observed before. Furthermore, even if a given relation has been observed frequently over some time interval, the distribution of occurrences over that interval might be highly inhomogeneous and bursty. Figure 1 shows that such non-stationarities exist for relations with different frequencies.

To alleviate this issue for static KGs, several recent studies have focused on adopting Few-Shot Learning (FSL) methods for KG reasoning under the condition that only a few triples (shots) are available for each relation. Xiong et al. 2018 and follow-up works [Chen et al., 2019, Wang et al., 2019a,b] aim to generate a similarity score to infer true entity pairs (subject, object) given the set of few training entity pairs for each relation. They obtain a representation for each entity pair through an encoder that aggregates the information from local neighborhood structure of each entity. However, FSL methods developed for static KGs are not adequate in temporal settings. Indeed, while data scarcity makes it even more imperative to consider temporal dependencies between events, existing static encoders designed for FSL methods [Xiong et al., 2018, Zhang et al., 2020, Du et al., 2019] are not able to incorporate such temporal dependencies that often contain crucial insights.

We address the above shortcomings of static FSL methods and propose One-shot Attention Temporal Graph Learning (OAT) for predicting new events for unseen relations and unseen timestamps in TKGs when there is only one training example for each relation. Our model can effectively encode an entity's interactions with others from $\ell$ previous times and capture the temporal dependencies between entities. It leverages a self-attention mechanism that sequentially aggregates an entity's neighborhood over time and extracts a time-aware representation for each entity. Furthermore, our model employs few-shot episodic training [Vinyals et al., 2016] to learn a similarity metric between a training entity pair and a given query entity pair, which indicates the likelihood of a given relation between the query entity pair. Unlike other existing TKG methods [Garcia-Duran et al., 2018, Jin et al., 2019], our model is able to predict future occurrences of a new relation type based on one training example, and without the need to fine-tune parameters to accommodate the new relation type. We also propose a new time-dependent approach to sample the training batch, and show its effectiveness over the random approach. Finally, we use past information to predict links in the future. Thus, it is capable of *extrapolation*, e.g., given the graph upto time $t_0$, inferring links for timestamps $t > t_0$, which is not addressed in previous TKG completion studies. This helps to populate the KG for future timestamps which could be of potential benefit for various applications.

Our contributions are as follows: (1) We formulate one-shot learning for TKGs, which improves upon existing low-shot techniques for static graphs, (2) We propose a temporal neighborhood encoder with a self-attention mechanism that effectively extracts the temporally-resolved neighborhood information for each entity, (3) We conduct experiments on two real-world datasets and demonstrate the superiority of the proposed model over state-of-the-art baselines, and (4) We construct two new publicly-available benchmarks for one-shot learning over TKGs.

## 2. Related Work

Our work is related to representation learning for temporal relational graphs, low-shot learning methods, and recent developments of meta-learning approaches for graphs.

**Low-Shot Learning.** An effective approach for low-shot learning is based on learning a similarity metric and a ranking function using training triples [Koch et al., 2015, Vinyals et al., 2016, Snell et al., 2017, Mishra et al., 2018]. Siamese networks [Koch et al., 2015] use a pairwise loss to learn a metric between input representations in an embedding space and then use the learned metric to perform nearest-neighbors separately. Matching networks [Vinyals et al., 2016] learn a function to embed input features in a low-dimensional space and then use cosine similarity in a kernel for classification. Prototypical networks [Snell et al., 2017] compute a prototype for each class in an embedding space and then classify an input using the distance to the prototypes in the embedding. SNAIL [Mishra et al., 2018] uses temporal convolution to aggregate information from past experiences and causal attention layers to select important information from past experiences. Another paradigm of low-shot learning includes optimization-based approaches that usually include a neural network to control and optimize the parameters of the main network. One example is MAML [Finn et al., 2017] which learns how to generalize with only a few examples for gradient updates.

**Relation Learning for TKGs.** TKG completion methods can be broadly categorized into translation-based methods, and evolving methods, based on their approach in encoding time information. The first category [Leblay and Chekol, 2018, Garcia-Duran et al., 2018, Dasgupta et al., 2018, Wang and Li, 2019, Jain et al., 2020] considers a distinct lower dimensional space such as a vector [Leblay and Chekol, 2018, Jain et al., 2020] or a hyperplane [Dasgupta et al., 2018, Wang and Li, 2019] for the event timestamps and defines a function to map an initial embedding to a time-aware embedding.

Evolving models assume a dynamic representation for entities or relations that is being updated over time. Such dynamics can be captured by a shallow encoder [Xu et al., 2019, Han et al., 2020a] or a sequential neural network [Trivedi et al., 2017, Jin et al., 2019, Wu et al., 2020, Zhu et al., 2020, Han et al., 2020b,c]. Xu et al. 2019 model the entities & relations as timeseries, and decompose the timeseries into three components using adaptive timeseries decomposition. DyERNIE [Han et al., 2020a] proposes a non-Euclidean embedding approach in the hyperbolic space. Trivedi et. al. 2017 represents events as the point processes, and Jin et. al. [Jin et al., 2019] aggregates the one-hop entity neighborhood at each timestamp by a pooling layer, and passes it to an RNN.

**Low-shot Learning for Graphs.** Metric based FSL models for graphs [Xiong et al., 2018, Chen et al., 2019, Zhang et al., 2020] adopt few-shot episodic training proposed by Vinyals et al. 2016 for few-shot link prediction over new unseen relations. In contrast, Beck et al. 2020 propose a FSL framework to address the unseen entities. These methods are usually composed of two components (i) an encoder that maps the support set to a low dimensional embedding, and (ii) a similarity network to compute the similarity score between the support set representation and a query.

Xiong et. al. 2018, Du et. al. 2019, and Zhang et. al. 2020 learn a representation for entities from their one-hop neighborhood. [Xiong et al., 2018] assumes that all the neighbors contribute equally, while Zhang et. al. 2020 assigns an attention weight to each neighbor. Sheng et.al. 2020 define a similarity score between a relation and the task relation and allows the neighborhood encoder to be dynamically adaptive to the task. MetaR [Chen et al., 2019] does not consider neighborhood interactions explicitly, and embeds a pair $(s, o)$ independently using $L$ fully connected layers. Qin et.al. 2020 studies zero-shot learning for relations using a generative adversarial network, where the generator tries to generate relations embeddings from text descriptions that are similar to the

real embeddings. [Wang et al., 2019b, Baek et al., 2020] focus on rare or unseen entities; Wang et. al. 2019b addresses this issue by incorporating entities textual description into the model. They further improve the performance by generating triples during training. Baek et.al. 2020 leverage the episodic training framework, but unlike previous methods, a task is associated with an entity. Unlike similarity-based approaches [Lv et al., 2019, Wang et al., 2019a] adopt the optimization-based meta-learning proposed in MAML [Finn et al., 2017]. The core of these approaches is to learn a parameter $\theta^*$ which can later be used as a good initialization for a new unseen task, and the model can be adapted to the new task with only a few gradient update steps. These approaches all assume a static graph. To the best of our knowledge, we are the first to study one-shot learning for TKGs.

## 3. Problem Formulation

In this section, we present the formal definition of the one-shot link prediction problem for TKGs.

### 3.1 Temporal Knowledge Graph Completion

A TKG is a collection of events represented as a set of quadruples $G = \{(s, r, o, t)|s, o \in \mathcal{E}, r \in \mathcal{R}\}$, where $\mathcal{E}$ and $\mathcal{R}$ are the set of entities and relations, and $t$ is the event timestamp. Static KG completion involves predicting new facts by inferring either a relation $r$ between two existing entities $s$ and $o$, or predicting the object entity $o$ given the subject entity $s$ and a relation $r$. In this work, we are interested in the later, but at a particular timestamp $t$. More formally, given an object query $(s, r, ?, t)$ and a set of candidates $\mathcal{C}$, the goal is to assign a likelihood to each entity, such that the true object entity is ranked higher than the other candidates in $\mathcal{C}$. The likelihood is estimated by a scoring function $\mathcal{P}$, usually parameterized by a neural network. We assume that the likelihood of an event depends not only on the current state of the graph, but also on the history of events from previous $\ell$ timesteps $\{t - \ell, \ldots, t - 2, t - 1\}$. In Section 4.1 we explain how to encode events' temporal information.

### 3.2 Few-shot Learning and Episodic Training

Generally, FSL focuses on building and training a model with only a few labeled instances for each class. Episodic training is a meta-learning framework for FSL proposed by Vinyals et al. 2016 and involves learning a model over a large set of tasks. Intuitively, each episode can be considered as a mini-training procedure with a training and a test set denoted as *support* and *query* set, respectively. At each iteration (episode), the training and test sets are sampled independently and an objective function $l_\theta$ is calculated over the test set. The goal is to learn a meta-model that maps a given (possibly unseen) support set to a model that performs effectively for the given query set. More specifically, assume we have a large set of tasks $\mathcal{T} = \{\mathcal{T}_i = (\mathcal{S}_i, \mathcal{Q}_i)\}_{i=1}^N$ where $\mathcal{S}_i$ and $\mathcal{Q}_i$ are the support and query set of task $\mathcal{T}_i$, the probabilistic optimization objective for this problem is given as:

$$\theta = \arg\max_\theta \mathbb{E}_{\mathcal{T}_i \sim \mathcal{T}} \Big[ \mathbb{E}_{\mathcal{S}_i \sim \mathcal{T}_i, \mathcal{Q}_i \sim \mathcal{T}_i} \Big[ l_\theta(\mathcal{Q}_i | \mathcal{S}_i) \Big] \Big]. \tag{1}$$

We adopt the standard episodic training framework in Eq. (1) for the purpose of TKG completion.

### 3.3 One-shot TKG Setup

As mentioned earlier, data scarcity is even a bigger problem in relational learning with TKGs. Few-shot episodic training has been proven to be effective to tackle this problem for static KGs [Xiong et al., 2018]. We further extend the framework proposed by Xiong et al. 2018 for TKG completion.

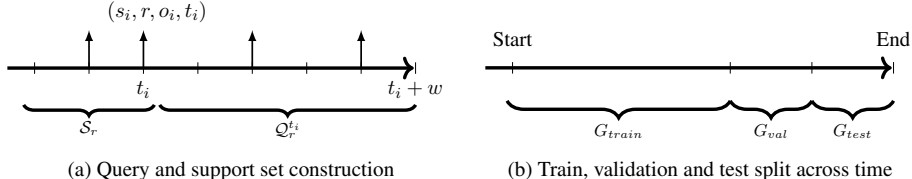

(a) Query and support set construction        (b) Train, validation and test split across time

Figure 2: (a) The query set is selected from $(t_i, t_i + w]$, with $t_i$ being the reference example timestamp in the support set(b) There is no time overlap between quadruples in validation and test.

Given a TKG, $G = \{(s, r, o, t)|s, o \in \mathcal{E}, r \in \mathcal{R}\}$, the relations in $\mathcal{R}$ are divided into two groups based on their frequency: frequent relations $\mathcal{F}$ and sparse relations $\mathcal{T}$. The sparse relations are used to build the task set needed by the model for the episodic training. Each task corresponds to a relation $r \in \mathcal{T}$ with its own training (support) and test (query) set, *i.e.* $\mathcal{T}_r = (\mathcal{S}_r, \mathcal{Q}_r)$ where $\mathcal{S}_r$ includes one reference entity pair at a specific time and $\mathcal{Q}_r$ is a set of query entity pairs, formally defined as:

$$\mathcal{S}_r = \{(s_0, o_0, t_0)|(s_0, r, o_0, t_0) \in G\}, \ \mathcal{Q}_r = \{(s_q, o_q, t_q)|(s_q, r, o_q, t_q) \in G\} \tag{2}$$

Given a relation $r \in \mathcal{T}$ and its reference set $\mathcal{S}_r$, one-shot TKG completion is to complete a given query $(s_q, r, o_q, t_q)$ where the object entity $o_q \in \mathcal{E}$ is missing. At each episode, a relation $r$ is selected randomly along with a quadruple containing that relation to form the support set. The query set can be selected in two ways: (i) **Random**: We randomly select $m$ positive quadruples containing $r$ from all the quadruples in $G$ (ii) **Time dependent**: The quadruples of the query set are restricted by their distance from the support set timestamp. More specifically, if the support quadruple timestamp is $\tau$ the time-dependent query set $\mathcal{Q}_r^\tau$ is defined as:

$$\mathcal{Q}_r^\tau = \{(s_q, r, o_q, t_q)|s_q, o_q \in \mathcal{E}, t_q \in \mathcal{I}_\tau\} \tag{3}$$

where $\mathcal{I}_\tau = (\tau, \tau + w]$ and the $w$ is called the *episode length*. Figure 2a illustrates the time constraint for selecting the query quadruples. Sampling procedure for the support and query set are provided in Section 4.3. The loss function $l_\theta$ at each episode optimizes a score function $\mathcal{P}_\theta$ such that it assigns a higher score to the true query events than the other events. $\mathcal{P}_\theta$ represents the similarity of a given query and the support set, and is proportional the event likelihood. The final optimization loss is:

$$\mathcal{L} = \mathbb{E}_{r \sim \mathcal{T}}\Big[\mathbb{E}_{\mathcal{Q}_r^\tau \sim G, \mathcal{S}_r^\tau \sim G}\Big[l_\theta(\mathcal{Q}_r^\tau|\mathcal{S}_r^\tau)\Big]\Big] \tag{4}$$

The relations in $\mathcal{T}$ are divided into mutually exclusive sets: $\mathcal{T}_{meta-train}, \mathcal{T}_{meta-test}, \mathcal{T}_{meta-val}$. From this, $G_{train}$ is defined as $G_{train} = \{(s, r, o, t)|r \in \mathcal{T}_{meta-train}\}$, with $G_{val}$ and $G_{test}$ defined similarly. We do not allow any time overlap between the quadruples in $G_{train}$, $G_{val}$ and $G_{test}$ since we are doing extrapolation and we do not want any information from the future timestamps to be given to the model. Figure 2b depicts the time split for $G_{train}$, $G_{val}$ and $G_{test}$. Finally, we assume that the model has access to a background knowledge graph defined as $G' = \{(s, r, o, t)|s, o \in \mathcal{E}, r \in \mathcal{F}\}$, and the entity set $\mathcal{E}$ is a closed set, i.e., there are no unseen entities during the inference time.

## 4. Model

Our key idea is to represent a relation $r$ by the entity pair in their support set $\mathcal{S}_r = \{(s_0, o_0, t_0)\}$. The similarity score $\mathcal{P}_\theta((s_q, o_q, t_q), S_r)$ between a query $(s_q, o_q, t_q)$ and the support set representation determines the likelihood of the event $(s_q, r, o_q, t_q)$ . For any task $\mathcal{T}_r$ (even unseen during the

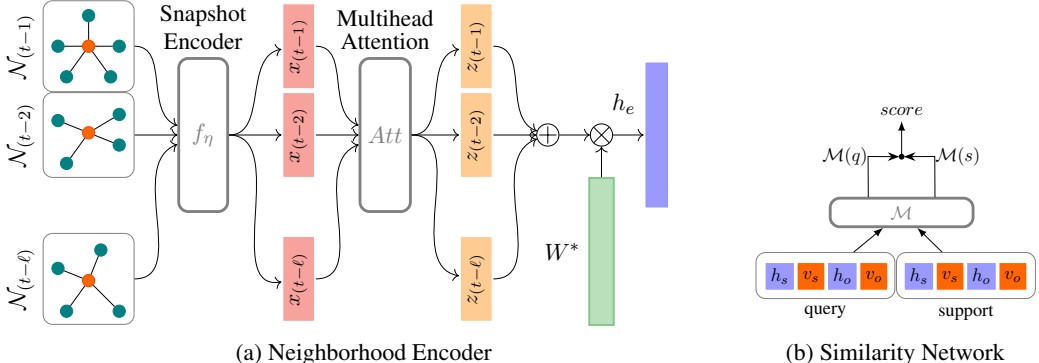

(a) Neighborhood Encoder  (b) Similarity Network

Figure 3: The main components of our model: (a) Temporal neighborhood encoder comprised of a snapshot encoder $f_\eta$ and the $Att$ modules. $\mathcal{N}_{(t-\ell)}, \ldots, \ldots, \mathcal{N}_{(t-2)}, \mathcal{N}_{(t-1)}$ represent the one-hob neighborhood of the node at time $(t-\ell)$ to $(t-1)$, given to the snapshot encoder $f_\eta$ as input. The $Att$ module produces a sequence of time-aware representation $z_{(t-\ell)}, \ldots, z_{(t-1)}$ aggregated by $W^*$ to obtain the final representation; (b) Similarity score is computed via the inner product.

training), this enables the model to predict the likelihood of a given query event for relation $r$, as long as one reference example $(s_0, o_0, t_0) \in \mathcal{S}_r$ is given to the model during the inference. To achieve the aforementioned goal, we propose a model that is built upon two steps: (i) obtaining the representation for the support set and query entity pairs, and (ii) computing the similarity of the support set and a query instance. Our model consists of two main components (Figure 3), as follows:

**Neighborhood Encoder**. The neighborhood encoder represents the neighborhood information of a given entity $e$ as a $d$-dimensional vector $h_e$. It encodes an entity's one-hop neighborhood in the graph during the previous $\ell$ timesteps as a sequence. In Section 4.1 we explain the detail of obtaining a test query and support set representation via the encoder.

**Similarity Network**. A similarity function parameterized by a neural network, $\mathcal{P}_\theta((s_q, o_q, t_q), S_r)$, that outputs a scalar similarity score between a query pair $(s_q, o_q, t_q)$, and the support set $S_r$. The similarity score indicates the likelihood of the query event $(s_q, r, o_q, t_q)$.

## 4.1 Neighborhood Encoder

For a given entity $e$, we define $\mathcal{N}_\tau(e) = \{(r_j, e_j)|(e, r_j, e_j, \tau) \in G'\}$ to be the one-hop neighborhood of $e$ at time $\tau$, and $e$'s temporal neighborhood $N(e)$ is the set of all $e$'s interactions within the previous $\ell$ timestamps, *i.e.* $\mathcal{N}(e) = \bigcup_{\tau \in [t-\ell, t-1]} \mathcal{N}_\tau(e)$. The neighborhood encoder is comprised of two parts: (i) function $f_\eta$ that encodes the one-hop neighborhood at a given timestamp $\tau$, and (ii) function $g$, that utilizes the output of function $f_\eta$, at previous timesteps, to generate a temporal neighborhood representation. Any pooling layer can be selected for $f_\eta$ and $g$, however, we show that the model can significantly benefit from the use of a sequential model as $g$.

### 4.1.1 SNAPSHOT AGGREGATION.

The snapshot aggregator $f_\eta$ aggregates local neighborhood information at a specific time $\tau$.

$$f_\eta(\mathcal{N}_\tau(e)) = \sigma(\frac{1}{C_{e_\tau}} \sum_{(r_j, e_j) \in \mathcal{N}_\tau(e)} (W^\top [v_{r_j} : v_{e_j}] + b)), \ x_\tau^e = [f_\eta(\mathcal{N}_\tau(e)) : v_e], \quad (5)$$

where $C_{e_\tau}$ is a normalizing factor, $v_{e_j} \in \mathbf{R}^{d \times 1}$, $v_{r_j} \in \mathbf{R}^{d \times 1}$ are entity and relation representations, and $W \in \mathcal{R}^{2d \times d}$ and $b \in \mathcal{R}^{d \times 1}$ are model parameters to be learnt. The concatenation is shown as $[:]$ and $\sigma(.)$ is a nonlinear activation function ($Relu$ in our implementation).

### 4.1.2 SEQUENTIAL AGGREGATION

Function $g$ aggregates the sequence of snapshots from previous $l$ timesteps $\{t - \ell, \dots, t - 2, t - 1\}$. To effectively capture the temporal dependencies between the timestamps, we use a sequential neural network. The sequential encoder employs the self-attention mechanism proposed in [Vaswani et al., 2017]. The core of the encoder is a layer denoted as *Att*, and is made up of two sublayers:
**Attention sublayer** projects the input sequence to a query and a set of key-value vectors.

$$\text{MultiHead}(Q, K, V) = [head_1 : \dots : head_h]W^O, \ head_i = \text{Attention}(QW_i^Q, KW_i^K, VW_i^V) \quad (6)$$

where $W_i^Q \in \mathbf{R}^{d_{model} \times d_k}$, $W_i^K \in \mathbf{R}^{d_{model} \times d_k}$, $W_i^V \in \mathbf{R}^{d_{model} \times d_v}$, $W^O \in \mathbf{R}^{hd_v \times d_{model}}$ are parameter matrices, and $d_{model}$ is the input embedding dimension ($d_{model} = 2d$ in our case).
**Position wise sublayer** is a fully connected feed-forward network, applied to each sequence position separately and identically.

The $Att(x, n_{head}, n_{layer})$ takes as input the neighborhood snapshots' representations $x = [x_{t-\ell}^e, \dots, x_{t-1}^e]$, the number of layers, and number of attention heads, and maps input sequence $x$ to a time-aware sequence output $z^e = Att(x, n_{head}, n_{layer})$, where $z^e = [z_{t-\ell}^e, \dots, z_{t-1}^e]$ and $z_\tau^e \in \mathbb{R}^{2d \times 1}, \forall t - \ell \le \tau \le t - 1$. Finally the temporal representation for $e$ at time $t$ is obtained by:

$$h_e = \sigma([z_{t-\ell}^e : \dots : z_{t-1}^e]W^*), \quad (7)$$

where $W^* \in \mathbf{R}^{2d \times d_{out}}$ is a parameter matrix, $[:]$ is concatenation and $\sigma(.)$ is a nonlinear activation function ($Relu$ in our implementation).

### 4.2 Similarity Network

Using the neighborhood encoder, any pair of subject and object at a given time $(s, o, t)$ can be represented with a vector $[h_s : v_s : h_o : v_o]$, where $h_s$ and $h_o$ are the temporal representations obtained from the neighborhood encoder, and $v_s$ and $v_o$ are the subject and object embeddings. Given the reference entity pair $(s_0, o_0, t_0)$ for a relation $r$, we learn the representation for similarity from the support and the query entity pair by two layers of fully connected layers:

$$x^{(1)} = \sigma(W^{(1)}x + b^{(1)}), x^{(2)} = W^{(2)}x^{(1)} + b^{(2)}, \mathcal{M}(x) = x^{(2)} + x. \quad (8)$$

The similarity score between the reference entity pair and a given query entity pair $(s_q, o_q, t_q)$ defined as $score = \mathcal{M}(s)^\top \mathcal{M}(q)$, where $s = [h_{s_0} : v_{s_0} : h_{o_0} : v_{o_0}]$ and $q = [h_{s_q} : v_{s_q} : h_{o_q} : v_{o_q}]$. We use the dot product to output a similarity score between the support and query pair that corresponds to the likelihood of $s_q$ and $o_q$ being connected with $r$ at time $t_q$.

### 4.3 Loss Function and Training

For a given relation $r$ and its support set $\mathcal{S}_r = \{(s_0, o_0, t_0)\}$, we have a set of positive quadruples $(\mathcal{Q}_r^+)$ and construct the negative pairs $(\mathcal{Q}_r^-)$ by polluting the subject or object entities for each positive quadruple and the final query set is $\mathcal{Q}_r = \mathcal{Q}_r^- \bigcup \mathcal{Q}_r^+$. We want the positive quadruples to be close to

---

**Algorithm 1**

| |
|---|

**Input:** $\mathcal{T}$ (meta training relations); $G'$ (background TKG); $w$ (episode length); $n_{shots}$ (number of shots); $\forall r, A_r = \{(s_i, r, o_i, t_i)\}$ ($t_i$ are sorted);

**for** $i = 1, 2, \ldots N$ **do**
    Shuffle relations in $\mathcal{T}$
    Sample relation $r$ from $\mathcal{T}$
    $\mathcal{S}_r, \mathcal{Q}_r \leftarrow \text{MAKETASK}(r, w, A_r, n_{shots})$
    Sample $B^+$ from $\mathcal{Q}_r$ and make $B^-$
    $\mathcal{L} \leftarrow \max(score^- - score^+\lambda, 0)$
    $\theta \leftarrow \theta - \nabla\mathcal{L}$
**return** $\theta$

**function** MAKETASK($r, w, A_r, n_{shots}$)
    $i \sim \text{Uniform}(1, |A_r|)$
    $\mathcal{S}_r \leftarrow \{A_r[i]\}$
    $limit \leftarrow w + t_i$
    **while** $t_j < limit$ **do**
        Add $A_r[j]$ to $\mathcal{Q}_r$
        $j \leftarrow j + 1$
    **return** $\mathcal{S}_r, \mathcal{Q}_r$

---

the support set's final representation and the negatives to be as far as possible. The objective function is a hinge loss, defined as $\mathcal{L} = max(score^- - score^+ + \lambda, 0)$.

The $score^+$ and $score^-$ are similarity scores calculated over $\mathcal{Q}_r^+$ and $\mathcal{Q}_r^-$. We employ episodic training over the task set $\mathcal{T}$ to optimize the loss function. Algorithm 1 summarizes the the episodic training algorithm and the time dependent selection to construct the query set.

## 5. Experimental Validation

We evaluate our model on predicting new events for a relation by predicting the object entity $(s, r, ?, t)$ and conduct qualitative and quantitative experiments to validate the model.

### 5.1 Datasets

We use two datasets: Integrated Crisis Early Warning System (ICEWS) [Boschee et al., 2015] and Global Database of Events, Language, and Tone (GDELT) [Leetaru and Schrodt, 2013]. From ICEWS dataset, we construct ICEWS17 from Jan 2017 to Jan 2019, and ICEWS14 from Jan 2014 to Jan 2016. We select the relations with frequency between 50 and 500 for the one-shot learning tasks and frequency higher than 500 as the background relations. Our second dataset includes one month of GDELT (Jan 2018). The low and high frequency thresholds for selecting tasks and background relations are 50 and 700, respectively, for the GDELT dataset. The rest of the dataset pre-processing is the same for both GDELT and ICEWS. Table 1 shows the statistics for both datasets.

| Dataset | # Ents | # Rels | # Tasks | # Quads |
|---|---|---|---|---|
| ICEWS14 | 3735 | 196 | 66/5/15 | 9793 |
| ICEWS17 | 2419 | 153 | 66/5/14 | 7535 |
| GDELT | 1549 | 204 | 50/5/14 | 10420 |

Table 1: Dataset statistics for ICEWS17(2017-2019), ICEWS14(2014-2016) and one month of GDELT (Jan 2018). #Rels include all the meta relations and background relations, and #Tasks is the number of relations in $\mathcal{T}_{meta-train}/\mathcal{T}_{meta-val}/\mathcal{T}_{meta-test}$.

### 5.2 Baselines

There is no prior work on one-shot learning for temporal knowledge graphs. Therefore, we propose two different ways to evaluate our model:

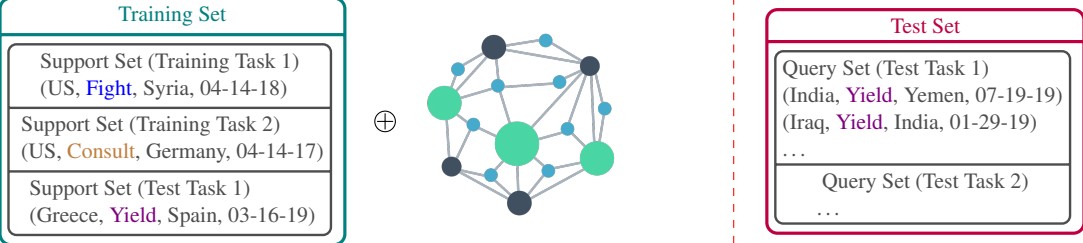

**Training Task 1**

Support Set
(US, Fight, Syria, 04-14-18)

Query Set
(Turkey, Fight, Iraq, 07-04-18)
(Israel, Fight, Syria, 05-01-18)
...

**Training Task 2**

Support Set
(US, Consult, Germany, 04-14-17)

Query Set
(India, Consult, Syria, 05-25-18)
(China, Consult, Japan, 01-01-19)
...

**Test Task 1**

Support Set
(Greece, Yield, Spain, 03-16-19)

Query Set
(India, Yield, Yemen, 07-19-19)
(Iraq, Yield, India, 01-29-19)
...

(a) Episodic training dataset. Each training/test task is associated with a relation. Test tasks contain relations and timestamps never seen during the training.

**Training Set**

Support Set (Training Task 1)
(US, Fight, Syria, 04-14-18)

Support Set (Training Task 2)
(US, Consult, Germany, 04-14-17)

Support Set (Test Task 1)
(Greece, Yield, Spain, 03-16-19)

⊕

**Test Set**

Query Set (Test Task 1)
(India, Yield, Yemen, 07-19-19)
(Iraq, Yield, India, 01-29-19)
...

Query Set (Test Task 2)
...

(b) Regular training dataset under one-shot condition. Training set contains the quadruples of the support sets from **all** the tasks plus the quadruples of the background knowledge graph $G'$. Test set include the quadruples of the query sets from the **test** tasks.

Figure 4: A schematic view of datasets for few-shot methods (a) and regular temporal methods (b).

1. **One-shot training of existing TKG models** To simulate the one-shot condition, we build a training dataset by adding all the quadruples of the background knowledge graph and the quadruples of the $G_{train}$. Per each relation in the $\mathcal{T}_{meta-test}$ and $\mathcal{T}_{meta-val}$, we include exactly one quadruple into the training set. We test the model on the exact same quadruples from $G_{test}$ (Figure 4) . We used TKG reasoning baseline models: TADistMult [Garcia-Duran et al., 2018], TTransE [Leblay and Chekol, 2018], ReNet [Jin et al., 2019], and ATiSE [Xu et al., 2019].

2. **FSL methods for static graphs:** We collapse the temporal training graph into an unweighted static graph. An edge exists between two entities in the static graph if there is a corresponding edge in the temporal graph at any time. We use three state of the art static low-shot learning methods: GMatch [Xiong et al., 2018], FSRL [Zhang et al., 2020], and MetaR [Chen et al., 2019]. Unlike the first two, MetaR doesn't incorporate any neighborhood information into its modeling, meaning that there is no difference between $(s, r, o, t_i)$ and $(s, r, o, t_j)$ in the test. In contrast, the one-hop neighborhood information provided for $(s, r, o, t_i)$ is different than $(s, r, o, t_j)$ during the test time of GMatch and FSRL.

**Discussion**. We present the results of our experiments in Table 2, which demonstrates the superiority of our approach over the baselines. In particular, for GDELT and ICEWS17 datasets, our model shows the best results according to all metrics, and at times with quite significant margins. For the ICEWS14 dataset, our model significantly outperforms the baselines according to the Hit@1 and MRR metrics, and is very close to the best methods in Hit@5 and Hit@10 metrics (TATransE and ReNet, respectively). We believe the superior performance of our model can be attributed to the episodic training approach, which provides more generalizability compared to the first set of baselines that use regular training. Our experiments confirm that the performance of TKG-inspired

| | GDELT | | | | ICEWS17 | | | | ICEWS14 | | | |
|---|---|---|---|---|---|---|---|---|---|---|---|---|
| Model | H@1 | H@5 | H@10 | MRR | H@1 | H@5 | H@10 | MRR | H@1 | H@5 | H@10 | MRR |
| TTransE | .025 | .075 | .138 | .060 | .004 | .047 | .107 | .038 | .004 | .076 | .134 | .058 |
| TATransE | .062 | .200 | .362 | .151 | .084 | .238 | .418 | .168 | .000 | **.377** | .489 | .175 |
| ATiSE | .059 | .195 | .297 | .138 | .064 | .325 | .456 | .196 | .031 | .248 | .357 | .137 |
| ReNet | .064 | .191 | .319 | .146 | .126 | .289 | .407 | .209 | .000 | .339 | **.542** | .164 |
| GMatch | .007 | .037 | .067 | .028 | .062 | .156 | .233 | .113 | .016 | .087 | .142 | .057 |
| FSRL | .080 | .158 | .210 | .127 | .120 | .253 | .345 | .192 | .039 | .095 | .153 | .074 |
| MetaR | .003 | .235 | .293 | .115 | .044 | .172 | .244 | .112 | .067 | .292 | .421 | .178 |
| OAT-R | .228 | .416 | .525 | .331 | **.191** | .479 | .641 | **.325** | **.164** | .330 | .538 | **.268** |
| OAT-T | **.234** | **.441** | **.578** | **.345** | .170 | **.519** | **.743** | .323 | .084 | .223 | .429 | .177 |

Table 2: Hit@K results for one-shot learning on (i) one month of GDELT (Jan 2018) (ii) ICEWS17 from Jan 2017 to Jan 2019), and (iii) ICEWS14 from Jan 2014 to Jan 2016 for relations in $\mathcal{T}_{meta-test}$. "OAT-R" and "OAT-T" correspond to the random and time dependent query set selection method.

baselines are better for frequent relations and deteriorate when evaluated over only sparse relations. TTransE/TATransE are translation based models which are not able to handle one-to-many/many-to-one relations, and in general perform poorly on event datasets. Similar to our model, ReNet generates a time-aware representation for an entity by aggregating the local neighborhood at each timestamp using a pooling layer and a sequential encoder. In Section B of the Appendix we provide some insight on why and when the ReNet model outperforms our model.

Although the second set of baselines employ episodic training, these methods still fail to consider the temporal dependency between events, which is captured effectively by self-attention in our model. GMatch uses a mean pooling layer to aggregate the entities and edges adjacent to the given entity, while FSRL uses a weighted mean pooling layer with attention weights. The reason that FSRL works better than GMatch might be that a part of the temporal information is captured by attention weights. To summarize, our experiments indicate that combination of both techniques – a self-attention to encode the temporal neighborhood information and a temporal task definition for episodic training – is essential for showing improved performance over the baselines.

## 5.3 Ablation Study

To demonstrate the importance of each component of our model, we conduct multiple ablation studies that evaluate the model from three main angles: (1) The temporal neighborhood encoder added by self-attention to the model: We disable the sequential encoder and feed all the neighbors of an entity in $\{t - \ell, \ldots, t - 1\}$ to the snapshot function $f_\eta$, as if they all happened at one timestamp (M1), shown in Table 3 as "Att". (2) Query set selection method: According to Section 3.3, it can either be random or time dependent. "Rand" is checked in Table 3 if the selection method is random, and time dependent otherwise. (3) We analyze the inner product effectiveness compared to using Matching Network [Xiong et al., 2018] on the query representation, shown as "MatchNet" in Table 3.

Table 3 summarizes the ablation study's results, showing that the full pipeline of our proposed algorithm outperforms the other variations. M1 and M2 demonstrate the effectiveness of a sequential encoder, since disabling it results a significant performance decline. It is worth noting that adding MatchNet helps to capture similarity information when the model is simple (M1 and M2). However,

| | Setting | | | GDELT | | | ICEWS17 | | |
|---|---|---|---|---|---|---|---|---|---|
| Model | Att | Rand | MatchNet | H@1 | H@10 | MRR | H@1 | H@10 | MRR |
| M1 | ✗ | ✗ | ✗ | .045 | .225 | .114 | .060 | .558 | .197 |
| M2 | ✗ | ✗ | ✓ | .133 | .504 | .243 | .105 | .518 | .220 |
| M3 | ✓ | ✓ | ✓ | .197 | .535 | .293 | .123 | .616 | .245 |
| M4 | ✓ | ✗ | ✓ | .169 | .491 | .265 | .138 | .654 | .269 |
| OAT-R | ✓ | ✓ | ✗ | .228 | .525 | .331 | **.191** | .641 | **.325** |
| OAT-T | ✓ | ✗ | ✗ | **.234** | **.578** | **.345** | .170 | **.743** | .323 |

Table 3: Ablation study on different components of the model (i) one month of GDELT (Jan 2018) and (ii) two years of ICEWS (Jan 2017 - Jan 2019) for relations in $\mathcal{T}_{meta-test}$.

comparing OAT-T/OAT-R with M3 and M4 shows that, due to the data scarcity, adding MatchNet will lead to model overparameterization and decreased performance, while self-attention is powerful enough to learn a representation that captures not only the temporal dependencies but also a similarity space that enables accurate prediction.

### 5.4 Performance Over Different Relations

We conduct experiments to evaluate the model performance for each relation separately. Table 4 in Section B of the appendix shows the decomposed results on ICEWS17 test set by OAT-T and ReNet, sorted by their frequency. It shows that in most relations (11 out of 14) OAT-T outperforms ReNet. Section B of the appendix include further discussion and detail of the experiments.

## 6. Conclusion and Future Work

We introduce a novel one-shot learning framework for temporal knowledge graphs to address the problem of sparse relations in those graphs. Our model employs a self-attention mechanism to sequentially encode temporal dependencies among the entities, as well as a similarity network for assessing the similarity between a query and an example. Our experiments demonstrate that the proposed method outperforms existing state-of-the-art baselines in predicting new events for infrequent relations. In future work, we would like to generalize the current one-shot learning to a few-shot scenario. Another direction is extending our framework to handle emergent entities, a challenge since new entities will have fewer interactions and thus sparser neighborhood information.

## 7. Acknowledgments

This research is based upon work supported in part by the Office of the Director of National Intelligence (ODNI), Intelligence Advanced Research Projects Activity (IARPA), via IARPA Contract No.2017-17071900005. The views and conclusions contained herein are those of the authors and should not be interpreted as necessarily representing the official policies, either expressed or implied, of ODNI, IARPA, or the U.S. Government. The U.S. Government is authorized to reproduce and distribute reprints for governmental purposes notwithstanding any copyright annotation therein.

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

## Appendix A. Attention Encoder Details

The Attention function used in Equation 6 to calculate the attention score is called "Scaled Dot Product Attention" in [Vaswani et al., 2017] and defined as follows:

$$\text{Attention}(Q, K, V) = softmax\Big(\frac{QK^T}{\sqrt{d_k}}\Big)V \tag{9}$$

**Position wise sublayer** is a fully connected feed-forward network, applied to each sequence position separately and identically.

$$FFN(x_\tau) = \max(0, x_\tau W_1 + b_1)W_2 + b_2. \tag{10}$$

In order for the attention model to make use of sequential order, a positional encoding is added to the input embeddings.

$$PE_{(pos,2i)} = sin(pos/10000^{2i/d_{model}})$$
$$PE_{(pos,2i+1)} = cos(pos/10000^{2i/d_{model}}), \tag{11}$$

Where $pos$ is the position and $i$ is the dimension. The purpose of positional encoding is to introduce to the model the information about relative or absolute position of each element in the sequence. The positional encoding has similar dimension as $d_{model}$.

## Appendix B. Performance Over Different Relations

In this section, we conduct experiments to evaluate the model performance over each relation separately. Table 4 shows relations in ICEWS test set by performance. Our model struggles on CAMEO codes "1831" and "1823," which lie under a higher level CAMEO event *"Assault"* coded as "18." Also, we manually inspected the test examples for "1823," for which ReNet performs very well. Our inspection shows that ReNet tends to generate higher ranks for a quadruple $(s, r, o, t)$ if it has already seen many examples of $s$, $o$ being paired with any other relations. For example, *(ISIS, 1831, Afghanistan)* was the test example, and we found 30 matches for *(ISIS, 183, Afghanistan)* in the training set. It is worth noting that "1831" is a subcategory of "183" in the CAMEO-code scheme. This was the case for 4 out of 5 query examples of "1831." The one query example that ReNet doesn't perform well, *(ISIS, 1831, Libya)*, the combination of $ISIS$ and any other $Libya$ related entities only appeared 7 times in the training data. The rank predicted by our model for this query is 14, while the ReNet rank is over 1,000. Our model doesn't use the information from the edges in the background graph. Although ReNet leverages this information, it could become biased toward them. Therefore, designing a few-shot model that leverages this information and is able to generalize well over new edges remains a challenge for future work.

## Appendix C. Performance over Time

Figure 5 visualizes the performance of our model over time for ICEWS dataset. Since relations selected for the task are very sparse, the number of query examples in one unit of time is very small. So we aggregated every 7 days. The y axis is the time difference between the query timestamp and its support example timestamp. Figure 5 shows that our model outperforms the best baseline over time.

| CAMEO Code | Description | Frequency | Hit@10 | |
| --- | --- | --- | --- | --- |
| | | | OAT-T | ReNet |
| 1044 | Demand change in institutions, regime | 52 | .300 | **.600** |
| 1125 | Accuse of espionage, treason | 58 | **.650** | .615 |
| 1311 | Threaten to reduce or stop aid | 64 | **.567** | .000 |
| 186 | Assassinate | 71 | **.533** | .250 |
| 1831 | Carry out suicide bombing | 97 | .450 | **.800** |
| 1122 | Accuse of human rights abuses | 121 | **.757** | .667 |
| 011 | Decline comment | 128 | **.600** | .000 |
| 0313 | Express intent to cooperate on judicial matters) | 130 | **.656** | .579 |
| 1823 | Kill by physical assault | 143 | .133 | **.286** |
| 1721 | Impose restrictions on political freedoms | 225 | **.600** | .133 |
| 0312 | Express intent to cooperate militarily | 273 | **.800** | .714 |
| 063 | Engage in judicial cooperation | 283 | **.688** | .364 |
| 0333 | Express intent to provide humanitarian aid | 292 | **.612** | .353 |
| 0332 | Express intent to provide military aid | 348 | **.785** | .463 |

Table 4: Hit@10 reported separately for each relation in the test tasks for ReNet and OAT.

## Appendix D. Hyperparameters and Implementation Detail

We select the relations with frequency between 50 and 500 for the one-shot learning tasks and frequency higher than 500 as the background relations for ICEWS dataset. The low and high frequency thresholds for selecting tasks and background relations are 50 and 700, respectively, for the GDELT dataset.The threshold for choosing the sparse relations should be selected such that the sparsity is preserved and also, we have enough data for training the model. We have selected the exact threshold values based on the prior work [Xiong et al., 2018] which also is based on the above rationale. GDELT is less sparse than ICEWS, so we increased the upper threshold to increase the number of tasks for the training.

We use a manual tuning approach to select the model hyperparameters, during which we keep all the parameters constant except one and we run the model with the selected hyperparameters 5 times and select the best model over the validation set using MRR metric.

The episode length $w$ chosen to construct the datasets for one-shot learning from GDELT and ICEWS is 120 time units (e.g. 120 days for ICEWS). The history period is 20 days for ICEWS and 10 time units (every 15 minutes) for GDELT. The embedding size for both datasets is 50. We use one layer of multi-head attention with 4 heads. Number of heads is selected by hyperparameter search from 1 to 6. Attention inner dimension is 256. Attention parameters are similar for both datasets. The matching network performs 3 steps of matching. The loss margin is 10 for ICEWS and 18 for GDELT. We found out that increasing margin value affects the performance as it is depicted in Figure 6. The number of parameters for the model with this choice of hyperparameters is 1,380,656 for GDELT dataset and 1,469,056 for ICEWS. We use Adam optimizer with initial learning rate 0.001.

We implemented our solution using Pytorch. We run all the experiments on a CPU `Intel(R) Xeon(R) Gold 5220 CPU @ 2.20GHz`, and 53 GBs of memory. The `eval` function in `trainer.py` includes the details to calculate MRR and Hit@K metrics. The implementation and the dataset is available at https://github.com/AnonymousForReview

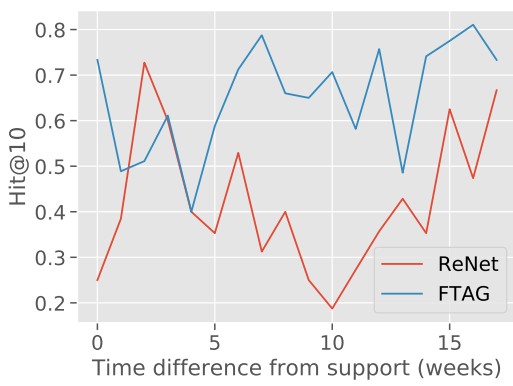

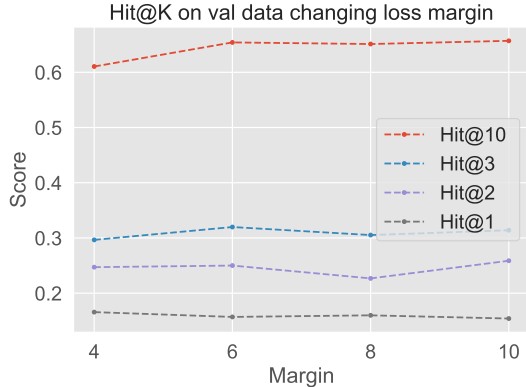

Figure 5: Model performance as the time between the prediction and the end date of the support set increases.

Figure 6: Model performance vs. the margin parameter.

## Appendix E. Model Analysis

In this section we provide more insights on the shortcoming of the existing baselines and the justification about why our model outperforms these models. We compare our model against two categories of baselines:

**TKG baselines**: Regular TKG methods tend to get biased toward the frequent relations. We conducted some initial experiments to confirm it; We provided a training set containing all the relations to the model, and evaluated it on all the relations as well as sparse/frequent relations separately. The model performance (MRR/Hit@K) over all the relations was more close to the MRR/Hit@K for frequent relations and the MRR/Hit@K for sparse relations was much lower. The main difference between regular TKG models and our model is the episodic training framework, which enables our model to generalize well from only one example.

- **TTransE/TATransE** are translation based models which are not able to handle one-to-many/many-to-one relations. they map the timestamp in a quadruple (s, r, o, t) into a lower dimensional space and are not capable of extrapolation (i.e. forecasting the future events).

- **ReNet**: Same as our model, ReNet generates a time-aware representation for an entity by aggregating the local neighborhood at each timestamp using a pooling layer and feeding it to an RNN. In Section B we provide some insight on why and when the ReNet model outperforms our model.

**FSL baselines**: The main difference between our model and static FSL models is a temporal neighborhood aggregator. Temporal adjacent events could convey useful information about the events that will happen in the future and different timestamps can have different effects on future events. The multi-head self-attention module in our model captures this information. We did some experiments on history length that indicated that as we increased the history length upto some point, it helped to improve the model performance.

- **GMatching** uses a mean pooling layer to aggregate the entities and edges adjacent to the given entity.

- **FSRL** uses a weighted mean pooling layer with attention weights. The reason that FSRL works better than GMatching might be that a part of the temporal information is captured by attention weights.

- **MetaR** does not use the local neighborhood structure for extracting the embedding of a node.

To summarize, our model combines the benefits of both approaches: a self-attention to encode the temporal neighborhood information and a temporal task definition for episodic training, resulting in better performance over the baselines.

## Appendix F. Data Construction

We provide the details of two newly constructed baselines for one-shot learning over temporal knowledge graphs. We conducted the following steps over both GDELT and ICEWS dataset:

1. A pre-processing step to deduplicate the dataset records by `Source Name` (subject), `Target Name` (object), `CAMEO Code` (relation), and `Event Date` (timestamp).

2. We divide the relations into two groups: frequent and sparse by their frequency of occurrence in the main dataset. Relations occurring between 50 and 500 in ICEWS, and 70 and 700 for GDELT are considered "sparse." Those occurring more than 500 times in ICEWS and more than 700 times in GDELT are considered frequent.

3. The quadruples of the main dataset are then split into two groups based on their relations. The quadruples containing frequent relations make background knowledge graph kept in `pretrain.csv`, and the quadruples containing sparse relations are kept for meta learning process (meta quadruples) kept in `fewshot.txt`

4. From the sparse relations, 5 are selected for meta-validation, 15 for meta-test and rest kept for meta-training.

5. We split the meta quadruples into meta-train, meta-validation, and meta-test not only based on their relations, but also based on the non-overlapping time split explained in Figure 2b of the paper.

### Data Format Description

Each constructed dataset contains the following files:

- **symbols2id.pkl**. A dictionary containing `ent2id`, `rel2id`, and `dt2id`, which are the mapping from entities, relations and dates to IDs respectively.

- **id2symbol.pkl**. A reverse mapping from IDs to symbols.

- **data2id.csv**. A file containing all the quadruples after the deduplication step. The symbols are represented by their ids.

- **pretrain.csv**. Contains the quadruples of the background knowledge graph.

- **fewshot.txt**. Contains the meta quadruples in text format. Each line is a tab separated quadruple with the order $s, r, o, t$.

- **meta_train.pkl**. A mapping from relations to meta quadruple IDs containing that relation. A quadruple ID indicates the line number corresponding to that quadruple in **fewshot.txt**. **meta_test.pkl** and **meta_val.pkl** are also created similarly, using meta-validation and meta-test relations.

- **hist_l_n**. A folder containing the entities' neighborhood information, with a maximum of $n$ neighbors at each snapshot and history length $l$. It includes the following files:

    - **hist_o.pkl**. The object neighborhood of meta quadruples in the **fewshot.txt**. The $i_{th}$ record corresponds to the quadruple in $i_{th}$ line of **fewshot.txt**.
    - **hist_s.pkl**. The subject neighborhood of meta quadruples in the **fewshot.txt**. The $i_{th}$ record corresponds to the quadruple in $i_{th}$ line of **fewshot.txt**.

.

