# OpenReview forum: "One-shot Learning for Temporal Knowledge Graphs"
_AKBC.ws/2021/Conference — AKBC 2021_

### Official Review · Reviewer_PPaE · 2021-07-21
**The paper introduces a novel task and benchmark datasets, but lack of comparisons with several recent baseline methods**

**Rating:** 5
**Confidence:** 4

**Review:**

The paper first introduces the task of one-shot learning for temporal knowledge graph and constructs two benchmark datasets to conduct experiments and make comparisons. For addressing the problem, the author proposes One-shot Attention Temporal Graph Learning (OAT) which utilizes self-attention to encode temporal interactions and a scorer network for link predictions. The experimental results and ablation study confirm the superiority of the proposed OAT approach against several TKG and FSL baselines.

Strengths:
1) The paper is generally well-written and easy to follow, the code and data are both provided for reproducibility.
2) This paper is well-motivated. One-shot learning for temporal knowledge graphs is an interesting research topic with practical scenarios and a wide range of applications. Two new publicly available benchmarks are also introduced by this paper and will benefit future research in this direction.
3) Compared with existing TKG models and few-shot learning methods for static graphs, the proposed OAT shows better performances on both GDELT and ICEWS benchmarks especially for sparse relations under the one-shot learning setting.

Weakness and Questions:
1) Although cited in the paper, the author forgets to compare with two recent competitive baselines in the experiment section. They are TeMP (Wu, Jiapeng, et al. “TeMP: temporal message passing for temporal knowledge graph completion”, EMNLP 2020) and ATiSE (Xu, Chengjin, et al. “Temporal knowledge graph embedding model based on additive time series decomposition”, arXiv preprint arXiv:1911.07893, 2019). Moreover, another approach called DySAT (Sankar, Aravind, et al. “Dysat: Deep neural representation learning on dynamic graphs via self-attention networks.” WSDM 2020) is also a recent competitive TKG model, but the author didn’t cite and compare it with DySAT in the paper.
2) TeMP and DySAT both apply self-attention to encode/integrate temporal information from the graph entity representations. The only difference is the similarity network of OAT, which is designed for the one-shot learning setting. Hence, the technical novelty of the paper is relatively limited.
3) The format (mainly the font and header) of the paper didn’t follow the original AKBC template and should be updated.

---

> ### Author Response · Authors · 2021-07-29
> **We ran ATiSE under one-shot setup and we will add more experiments if appropriate (more details in general response above)**
>
> Thank you for your comments and insightful feedback. We applied your comments and tried to improve the paper based on your feedback.
>
> You had raised points about the novelty and the limited experiments of the paper. Please see our detailed response in the general comment above. To summarize, we have added one more baseline (ATiSE) to the experiments based on your suggestion and will continue to work on adding TeMP to the revised version.
>
> We will make sure that the format is fixed in the revisions.

---

### Official Review · Reviewer_Bk3F · 2021-07-21
**A new setting for temporal knowledge graphs,**

**Rating:** 7
**Confidence:** 4

**Review:**

This paper proposes a new setting for temporal knowledge graphs ("TKG") --- one-shot link prediction for TKGs. The motivation is that knowledge graphs are long-tail (thus we need one-shot/few-shot link prediction) and for temporal knowledge graphs, the distribution of relational facts can be highly heterogeneous. The authors formalize the one-shot link prediction as an episodic learning problem, where in each episode there is a support set and a query set. The quadruples are divided into meta-train, meta-val and meta-test sets based on the timestamp. The authors also propose a new method to tackle the problem, including a neighborhood encoder to get entities' representation, with consideration of the graphs at previous timesteps, and a similarity network to get the similarity between query and support instances. The authors compare their model with several TKG or one-shot link prediction baselines on three datasets and show that the proposed model (OAT) achieves superior performance. The ablation study verifies the effectiveness of each component of the method.

Strength:

1. The paper is well motivated and the newly proposed setting is challenging and meaningful.
2. The proposed method is well described and is very intuitive.
3. The results are very strong and the ablation study is very comprehensive.

Weakness:

1. It would be nicer to provide some examples of relations and quadruples, especially for such a new setting.
2. The paper mainly talks about the one-shot setting. A few-shot setting and method would be more generalizable and more realistic in applications.
3. Although the authors have already compared several baselines, it would be better to compare the proposed method to more recent TKG baselines (it is even more necessary considering the paper is setting a new direction).

Question:

1. Do you have any idea why OAT-T is much worse than OAT-R on ICEWS14 where it is better on the other datasets?

Misc.:

1. Missing space on page 3: "Xiong et. al. 2018,Du et. al. 2019"
2. Eq. (1): isn't Ti~T and Si, Qi the same thing?

---

> ### Author Response · Authors · 2021-07-29
> **Further clarification and examples on model setting, and design, We also ran more baselines (details in general response above)**
>
> Thank you for your time and your valuable and encouraging comments, which we think will help to improve the paper. Please also see our general comment regarding the novelty and the experiments.
>
> **Examples of Support & Query Set:**  We have added examples and a pictorial demonstration on how the support set and the query set are created in meta training/validation/test for the episodic training. Since OpenReview does not allow figures in the response, here’s an example as structured text:
>
>
> Episodic Training :
> ```
>  Meta-Train : {
>      Training Task 1: {
> 		-Support set : {(US, Fight, Syria, 04-14-18)}
> 		-Query set : {
> 			(Turkey, Fight, Iraq, 07-04-18)
>             (Israel, Fight, Syria, 05-01-18)
>             (US, Fight, Syria, 04-14-18)
>          ....
> 		}}
> 	Training Task 2: {
> 		- Support set: {(US, Consult, Germany, 04-14-17)}
> 		- Query set: {
>            (India, Consult, Syria, 05-25-18)
>            (China, Consult, Japan, 01-01-19)
>            ....
> 		}}
>         ....
>   }
> ```
>
>
> ```
> Meta Test: {
> 	Test Task 1: {
> 		-  Support set : {(Greece, Yield, Spain, 03-16-19) }
> 		- Query set: {
>            (India, Yield, Yemen, 07-19-19)
>            (Iraq, Yield, India, 01-29-19)
>            (France, Yield, UK, 02-10-19)
>                            ....
> 		}}
>      .....
> }
> ```
>
>
> **Few-shot learning:**  We agree that the few-shot setup is more general but extension to this set is beyond the scope of our work. Please also note that the one-shot setup has practical implications, especially when the data is sparse and there is only one example. In addition, the episodic training framework introduced in the paper and the definition of support set and query set are naturally extendable from one-shot to few-shot setup.
>
> **Response to you question, OAT-T vs OAT-R:** Our intuition is that the more non-stationary the data is, the more OAT-T would be helpful. To be more specific, it would be difficult for the model to generalize from only one sample, to the quadruples that are selected from a period of three/two years of data, if the distribution changes drastically over the course of three years.
>
> **Typos & Misc:** Thank you for reading the paper carefully. We have corrected the typos.
> Eq. (1): isn't $T_i \sim T$ and $S_i$, $Q_i$ the same thing? This equation is meant to describe few-shot learning in general terms for classification tasks. In that case, $T_i \sim T$ consists of a set of labeled data, and at each iteration, $S_i \sim T_i$ and $Q_i \sim T_i$ are sampled to calculate the optimization objective. We have changed $S_i$, $Q_i$ to $S_i \sim T_i, Q_i \sim T_i$ in the paper.

---

### Official Review · Reviewer_Ljqx · 2021-07-23
**TKG Completion in one-shot setting**

**Rating:** 6
**Confidence:** 4

**Review:**

The work proposes a principled approach to knowledge completion in temporal knowledge graphs. This is achieved by two novelties — 1) a one-shot learning framework for TKGs that is based off similar approaches in static KGs and 2) a temporal neighborhood encoder that allows modeling of graphical and temporal dependencies, perhaps the key contribution of the work. Further, the work reports detailed exports highlighting the utility of their approach in addition to providing two additional benchmark datasets.

Comments and Questions:
- Caption under Fig 3 can be improved — e.g. explaining the notation / input / output.
- The paper is generally well-written and the described method is easy to follow
- The two architectural contributions proposed are not novel by themselves. However, their application in this setting is interesting and their utility, as demonstrated by experiments can be of use to the relevant community.
- Do the authors have intuition on the pros/cons of OAT-R or OAT-T for different types of data? Are there certain temporal properties that might favor one vs other? This might be useful to someone interested in applying the methodology for their problem.

---

> ### Author Response · Authors · 2021-07-29
> **More intuition on model design and architecture**
>
> Thank you for your time and valuable feedback.
>
> * We have updated the caption for Fig 3.
>
> * As you mentioned, the main purpose of the paper is to introduce a novel one-shot link prediction setting for temporal knowledge graphs as well as a novel time-dependent episodic training for TKGs. We empirically  show that the combination of a self-attention temporal encoder with episodic training would significantly enhance the link prediction performance
> performance for infrequent relations in temporal knowledge graphs.
>
>
> * **OAT-R vs OAT-T**: Our intuition is that, the more non-stationary the data is, the more OAT-T would be helpful. To be more specific, it would be difficult for the model to generalize from only one sample, to the quadruples that are selected from a period of three/two years of data, if the distribution changes drastically over the course of three years.

---

### Official Review · Reviewer_nKEs · 2021-07-24
**New Setting for Temporal Knowledge Graph Link Prediction, but with limited experiments**

**Rating:** 6
**Confidence:** 4

**Review:**

### Summary:

The paper addresses the problem of one-shot link prediction for temporal knowledge graphs (TKG). Specifically, it focuses on relations with very few triples. The proposed approach uses the framework of few-shot learning and episodic training on TKG using a temporal neighborhood encoder. The experiments demonstrate the proposed approach's effectiveness on three datasets against existing TKG models and few-shot learning models for static graphs.


### Strengths:
1. The paper presents a new setting for TKG link prediction. The paper also adapts the existing TKG datasets for the few-shot setting.
2. The proposed approach performs well across three datasets.
3. The problem is well-motivated, and the paper is well-written and easy to follow.

### Weaknesses:
1. Although it is a new setting, the temporal neighborhood encoder is similar to existing works (e.g., TeMP). Therefore, the novelty of the proposed method is limited. Further, it does not compare with alternate temporal encoders. Many TKG encoders are present in the literature, and it would have been interesting to evaluate their effectiveness in this setting.
2. The evaluation dataset only consists of relations with frequency >= 50. It would have been interesting to see the method's effectiveness on highly infrequent relations.


### Questions to the author:
- Did you also try relations with frequency <= 50? If yes, how did it perform?

### Presentation Improvements:
- The experiments section in the main paper is small. Probably, it can include concise versions of some experiments from the appendix.

---

> ### Author Response · Authors · 2021-07-29
> **Clarification on evaluation datasets. We ran more baselines; please see our general response above for more details.**
>
> Thank you for your detailed comments and insightful feedback. Please see our general comment in response to the points raised about the novelty and the experiments.
>
> **Evaluation Dataset:** The entire training/validation/test is done over the relations with a frequency between 50< < 500. Please note we have followed the setup in Xian et. al 2018. Relations with frequency > 500 are used as metadata and as you noted, the relations with frequency < 50 are ignored. The ٰreason for ignoring relations with frequency < 50  was that when the quadruples of a relation only appear 50 times in course of a two year period, it is highly likely that they appear less than 5 times during the test/validation period (3 months of data in our experiments), thus they wouldn’t have a significant effect on the final results.
>
> **Presentation:** We thank the reviewer for the suggestion and will provide a concise summary of experiments in the main body of the paper while keeping details in the appendix.

---

### Author Response · Authors · 2021-07-29
**General Response to All The Reviewers**

We thank the reviewers for their detailed comments and insightful feedback. Two repeating themes in the reviews were concerns about the limited novelty of our work and the insufficient number of baselines used in our experiments. Below we provide our response on these two issues. Other questions raised by the reviewers will be addressed via individual responses.

### **Novelty:**
We stress that the main novelty of our work is the formulation of the one-shot link prediction problem for temporal knowledge graphs and the introduction of a novel one-shot episodic learning for TKGs and. Furthermore, while the attention mechanism for temporal encoding was suggested before, empirically validating the effectiveness of self-attention for a one-shot setting is also an innovative aspect of our work.

### **Insufficiency of the baselines:**
First, We note that the baselines used in our work were state-of-the-art methods at the time of finalizing our experiments, e.g., ReNet had the best performance and many of the recent TKG methods had not been peer-reviewed or did not have a proper implementation to work with.

We now elaborate on specific methods mentioned by the reviewers:

* **ATiSE:** Following the suggestion by Reviewer, we have run ATiSE (Xu, Chengjin, et al. “Temporal knowledge graph embedding model based on additive time series decomposition”, arXiv preprint arXiv:1911.07893, 2019) on ICEWS14 and ICEWS17. Our preliminary results suggest that our method outperforms ATiSE in all metrics for both datasets. A summary of the results are as follows:

|       | ICEWS14 |       |       |       | ICEWS17 |       |       |       |
|-------|---------|-------|-------|-------|---------|-------|-------|-------|
|       | H@1     | H@5   | H@10  | MRR   | H@1     | H@5   | H@10  | MRR   |
| **ATiSE** | 0.064   | 0.325 | 0.456 | 0.196 | 0.031   | 0.248 | 0.357 | 0.137 |
| **OAT-R** | **0.164**   | **0.330** | **0.538** | **0.268** | **0.191**   | 0.479 | 0.641 | **0.325** |
| **OAT-T** | 0.084   | 0.323 | 0.429 | 0.177 | 0.170   | **0.519** | **0.743** | 0.323 |

Note: We have used the hyper-parameters used in ATiSE for ICEWS14, except for the embedding size that we set to 50 to be consistent with our model embedding size.

* **TeMP:** We did not compare our approach to TeMP because the latter considers interpolation setup for graph completion (smoothing), whereas our work targets extrapolation/forecasting setup, where one predicts links for future timestamps. We note that the authors of the TeMP paper themselves did not compare their approach to extrapolation-based methods such as ReNet, e.g.  in Section 4.3 of their paper, they state that:
_“We don’t compare with RE-NET, GHN (Han et al., 2020), DartNet (Garg et al., 2020) and Know-Evolve since these work focus on graph extrapolation task.”_
  We are currently trying to compare TeMP to our method but having issues running their code for the extrapolation setting. We will update our response if we are able to generate results in time. We also note that except for the self-attention mechanisms, TemP is very similar to RE-Net, so we don’t expect any significant difference in the results.

* **DySAT:** As you pointed out, DySAT uses a self-attention mechanism for temporal graphs, however, it is not directly applicable for temporal knowledge graphs with different relation types.  They do not conduct experiments on any of the common TKG benchmarks such as GDELT and ICEWS nor do they compare with any state-of-the-art TKG model. We have cited their papers in the related work as it can be related to the self-attention encoder of our model. However, we consider DySAT to be more related to the dynamic graphs literature.

---

### Decision · Program_Chairs · 2021-08-18

**Decision:**

Accept

**Comment:**

This paper addresses the problem of one-shot link prediction for temporal knowledge graphs (TKG), with a specific focus on sparse relations (ie with very few triples), using few-shot learning and episodic training on TKG with a temporal neighborhood encoder. While there were some questions around novelty and sufficiency of baselines, overall the paper presented very strong results in a very clear manner for a relevant and interesting problem.